# Possibilities of Real Time Monitoring of Micropollutants in Wastewater Using Laser-Induced Raman & Fluorescence Spectroscopy (LIRFS) and Artificial Intelligence (AI)

**DOI:** 10.3390/s22134668

**Published:** 2022-06-21

**Authors:** Claudia Post, Niklas Heyden, André Reinartz, Aaron Foerderer, Simon Bruelisauer, Volker Linnemann, William Hug, Florian Amann

**Affiliations:** 1Department of Engineering Geology and Hydrogeology, RWTH Aachen University, Lochnerstr. 4-20, 52064 Aachen, Germany; andre.reinartz@rwth-aachen.de (A.R.); amann@lih.rwth-aachen.de (F.A.); 2Institute of Geomechanics and Underground Technology, RWTH Aachen University, Mies-van-der-Rohe-Straße 1, 52074 Aachen, Germany; foerderer@gut.rwth-aachen.de; 3Independent Researcher, Altburgstrasse 45, 8105 Regensdorf, Switzerland; sbruelisauer@bluewin.ch; 4Institute of Environmental Engineering (ISA), RWTH Aachen University, Krefelder Str. 299, 52070 Aachen, Germany; linnemann@isa.rwth-aachen.de; 5Photon Systems Inc., 1512 Industrial Park St., Covina, CA 91722-3417, USA; w.hug@photonsystems.com

**Keywords:** environmental monitoring, micropollutants, data processing, real-time monitoring, wastewater treatment plant, DUV Raman/fluorescence spectroscopy, artificial intelligence

## Abstract

The entire water cycle is contaminated with largely undetected micropollutants, thus jeopardizing wastewater treatment. Currently, monitoring methods that are used by wastewater treatment plants (WWTP) are not able to detect these micropollutants, causing negative effects on aquatic ecosystems and human health. In our case study, we took collective samples around different treatment stages (aeration tank, membrane bioreactor, ozonation) of a WWTP and analyzed them via Deep-UV laser-induced Raman and fluorescence spectroscopy (LIRFS) in combination with a CNN-based AI support. This process allowed us to perform the spectra recognition of selected micropollutants and thus analyze their reliability. The results indicated that the combination of sensitive fluorescence measurements with very specific Raman measurements, supplemented with an artificial intelligence, lead to a high information gain for utilizing it as a monitoring purpose. Laser-induced Raman spectroscopy reaches detections limits of alert pharmaceuticals (carbamazepine, naproxen, tryptophan) in the range of a few µg/L; naproxen is detectable down to 1 × 10^−4^ mg/g. Furthermore, the monitoring of nitrate after biological treatment using Raman measurements and AI support showed a reliable assignment rate of over 95%. Applying the fluorescence technique seems to be a promising method in observing DOC changes in wastewater, leading to a correlation coefficient of R^2^ = 0.74 for all samples throughout the purification processes. The results also showed the influence of different extraction points in a cleaning stage; therefore, it would not be sensible to investigate them separately. Nevertheless, the interpretation suffers when many substances interact with one another and influence their optical behavior. In conclusion, the results that are presented in our paper elucidate the use of LIRFS in combination with AI support for online monitoring.

## 1. Introduction

One of today’s most pressing global challenges is the availability of a sufficient amount of drinking water of adequate quality. A major threat that is currently posed to surface and groundwater reservoirs is their contamination by micropollutants (pharmaceuticals, pesticides, industrial chemicals, consumer care products). The discharge of harmful substances into the aquatic environment via municipal wastewater treatment plants (WWTP) and combined sewer overflows contributes significantly to the pollution of water bodies [1,2]. Some of these harmful substances appear in µg/L or even ng/L concentrations which overwhelms the sensitivity of many analysis techniques. These so-called ‘emerging micropollutants’ are traces of medical products, plant protection products, biocides and other chemicals that can have detrimental effects on the environment or human health at very low concentrations [3,4]. In 2010, the largest ever EU-wide monitoring survey on WWTP effluents was performed. The obtained results show the presence of 131 target organic compounds in European wastewater effluents, in concentrations ranging from low nanograms to milligrams per liter [5]. Due to the adverse effects on human and environmental health, a legal obligation to reduce and prevent micropollutants is posed by the European Water Framework Directive (WFD) and the environmental quality standards (EQS) for priority substances [3,6,7]. For the substances that are defined in the daughter directive of the WFD the gradual reduction or even cessation of discharges and emissions is to take place [8].

In consequence, specific measures must be implemented to combat water pollution that is caused by relevant pollutants or pollutant groups as soon as environmental quality standards (EQS) are exceeded [9]. These specific measures presuppose that the nature and extent of micropollutants are recognized, for example, in WWTP and their relieving surface water bodies. Unfortunately, most micropollutants are not yet included in the routine monitoring of surface and wastewater [10]. This means that, compared to other anthropogenic contaminants the emerging micropollutants (EMPs) have largely fallen outside the scope of monitoring and worldwide regulations [11]. The occurrence of different organic micropollutants, especially EMPs (even at very low concentrations from point and nonpoint sources) in the natural environment has raised significant concerns about their negative effect on aquatic ecosystems and human health [12,13,14].

Municipal wastewater treatment plants were designed to treat domestic wastewater, not to reduce, for example, pharmaceuticals, pesticides and microplastics. Biologically, WWTP remove protein, fat and carbohydrates, as well as the important plant nutrients carbon, phosphorus and nitrogen from the water [15,16]. It must be considered that today’s conventional wastewater treatment technology (mechanical and biological steps) was designed in the 1970s [5]. This fact underlines to the problem that many substances are not even covered by the normal sum parameter monitoring of a WWTP, thus the chemical inventory of the wastewater is not known, meaning that it cannot be completely purified with the available treatment stages [17]. In fact, most of the conventional WWTP are not designed to eliminate organic compounds at low concentrations, making the treatment processes vulnerable and triggering a dissemination of micropollutants all over the environmental compounds (Figure 1) [18,19].

Information on the degradation of, for example, diclofenac (analgesic) at WWPT ranges from “no effect” to a 60% degradation rate for the different size classes and the different stages of expansion of the WWTP [20,21,22,23,24]. A review paper from Luo et al., 2014, showed that the WWTP removal efficiency for selected micropollutants in 14 countries/regions depicted compound-specific variation in removal, ranging from 12.5 to 100%. It must be noted that the consumption quantities of selected drugs in Germany that are the subject of public debate exceed 1000 t per year in some cases (e.g., for ibuprofen) [9]. In consequence, regarding pharmaceuticals, toxicological studies have shown that they might have direct toxicity towards certain aquatic organisms [25,26,27,28]. The occurrence of micropollutants in the aquatic environment has frequently been associated with several negative effects, including short-term and long-term toxicity, endocrine-disrupting effects and the antibiotic resistance of microorganisms [2,29,30]. Their constant but imperceptible effects can gradually accumulate, finally leading to irreversible changes in both wildlife and human beings [31,32]. The presence of these chemicals in the environment is more concerning considering that they do not appear individually, but as a complex mixture, which could lead to unwanted synergistic effects. The ubiquity of a high number of potentially toxic emerging chemicals in the environment underpins the need to better understand their occurrence, fate and ecological impact [33]. This leads to the recurring demand for online monitoring that follows the concentrations, retention time and chemical interactions between the relevant substances. Therefore, high demands on sensitivity, high throughput by automation and short analysis times are major requirements [34].

Our research tests the possibility of the real-time monitoring of micropollutants in wastewater using laser-induced Raman & fluorescence spectroscopy (LIRFS) in combination with an artificial intelligence approach using convolutional neural networks (CNN). Spectroscopy in general comprises a group of optical techniques that decompose radiation according to the properties of molecules, such as wavelength or energy, producing a spectrum as an excitation response. Radiation in the form of electromagnetic waves maps the specific properties of a medium between the light source and the spectrometer, producing a specific spectrum. The spectrum contains information to confirm atomic and molecular models, determine qualitative compositions, and ascertain the proportion of a substance in a mixture that is based on the spectra intensity [35,36]. In our current research case, we investigate both Raman and fluorescence spectroscopy that was conducted with excitation in the deep-UV below 250 nm on the influent and effluent wastewater of a biological treatment stage and an ozonation stage at wastewater treatment plants. We identified the substances that affect the purification processes in WWTP; influence the installed sum-parameter monitoring (e.g., humic acids); result from purification processes (as a transformation product like bromate); or pass the WWPT without substantial reduction (e.g., pharmaceuticals like diclofenac, carbamazepine), and may lead finally to a degradation of the surface water quality. Our approach aims to identify and quantify the problematic pure substances using LIRFS in combination with a CNN classification model. On every influent and effluent sample fluorescence and Raman measurements were performed and the reliability of the results were statistically evaluated and trained by CNN.

## 2. Fundamentals

### 2.1. Fluorescence/Raman Spectroscopy

Fluorescence spectroscopy is a method that is based on energy absorption, where a fluorophore molecule is excited into various excited states through the absorption of electromagnetic radiation of a certain wavelength. The radiation is emitted following the transition to the lowest energy state (Kasha rule) and has a longer wavelength (less energy) than the absorbed photon. The wavelength difference between the absorption maximum and the emission maximum is called the Stokes shift [37] (see Figure 2, left). The molecule emits the energy in the form of fluorescent light of a certain wavelength, in order to return from its excited state to the ground state (see Figure 2, right) [38,39].

The process of fluorescence can be described using the simplified Jablonski diagram in Figure 2 (right). In the Jablonski diagram, the energy levels of the ground state (S0) and various excited states (S1, S2, S3), as well as the vibrational levels are shown. A fluorophore molecule can become excited from the ground state (S0) through the absorption of a photon into a higher energy level (S1, S2, S3), indicated by the violet arrow. The excited electron of a fluorophore molecule returns to the lowest vibration state of the first excited state (S1) due to internal conversion (blue arrows). From the lowest vibration level, the electron falls to the ground state by emitting the energy due to fluorescence (green arrow).

Another optical method is Raman spectroscopy. The Raman effect can be described as a two-photon process. Raman scattering is observed when monochromatic light hits a molecule and deforms or polarizes the electron shell, creating a short-lived so-called ‘virtual state’ of the electron. The energy absorption of the incident photon is called anti-Stokes scattering, whereas the energy transfer to an emitting photon is called Stokes scattering [41,42].

As we mentioned in our previous paper [43], deep-UV spectroscopy offers the opportunity to analyze Raman and fluorescence emissions separately by using a nearly fluorescence-free wavelength range.

This was first demonstrated by Asher et al. [44]. As Liu [45] summarized, a wavelength range was discovered in which the interfering fluorescence can be separated from the Raman signals.

Photon Systems Inc. [46] showed a spectral bandwidth over 3000 1/cm in which no interference from fluorescence was observed by DUV-Raman excitation at a wavelength that was below λ = 250 nm. This effect is well shown by Photon Systems on the Raman and fluorescence spectra of crude oil that is excited by different wavelengths [47,48].

The fluorescence behavior and intensity can be significantly influenced by different chemical and physical factors, called quenching. Quenching in general refers to all processes that decrease the fluorescence intensity, such as reactions in the excited state, the transfer of energy, molecular collisions and complexation reactions. The term collision quenching refers to intermolecular interactions, e.g., in the form of shocks, by which the excited fluorophore returns to ground state after colliding with a quencher [39]. The mechanism of fluorescence quenching depends on the colliding molecule pair, especially on the substances which act as quenchers (in particular, oxygen, halogens, amines and molecules with electron deficiency). In addition to quenching, the internal filter effect and metal ions in the water influence the fluorescence behavior.

The fluorescence intensity is furthermore greatly influenced by temperature, whereby the correlation between these factors is anti-proportional. In the event of a temperature rise, the probability that an excited electron is caused by radiation-free decay increases. How great the effect on the fluorescence intensity is depends on the composition and size of the fluorophore [49].

The complexity of the largely unexplored behavior between the individual relationships and thus their influence on fluorescence is the subject of numerous works in the literature [50,51,52].

### 2.2. Wastewater Treatment and Monitoring

Wastewater treatment developed rapidly in the last decades due to ongoing challenges in wastewater composition. All the purification processes are undergoing a process of optimization, so this brief overview emphasizes a bundle of applied process combinations but cannot guarantee completeness.

The aim of the first treatment stage is to remove the undissolved components from the wastewater. In this process, coarse materials are retained by screens, sediments are forced to sediment by lowering the flow velocity, and floating light materials are removed by skimmer devices. A further, detailed explanation is given in [53,54,55]. In Gupta et al., 2012 [56], a good illustration of primary, secondary and tertiary treatments and water recycling technologies is shown.

Secondary water treatment includes biological routes for the removal of soluble and insoluble pollutants by microbes [57]. The microbes, usually bacterial and fungal strains, convert the organic matter in an aerobe process (nitrification) and later in an anaerobe process (denitrification) into methane and inorganic substances, such as carbon dioxide and ammonium [58]. Many microorganisms form colonies which become visible as sludge flocs. These sludge flocs settle at the bottom of the subsequent secondary settling pond or clarifier and are either returned to the activated sludge process or pumped into the primary settling pond for sludge disposal (see Figure 3). The biological treatment includes the aerobic and anaerobic digestion of wastewater. The aerobic process is effective for the removal of biological oxygen demand (BOD), chemical oxygen demand (COD), dissolved and suspended organics, volatile organics, organically bound nitrogen and ammonia, phosphates etc. The concentration of biodegradable organics can be reduced by up to 90% [56]. If free-dissolved oxygen is not available in the wastewater, anaerobic decomposition occurs. Anaerobic and facultative bacteria convert the complex organic matter into simpler organic compounds that are based on nitrogen, carbon and sulphur [58]. The important gases that are involved in this process are nitrogen, ammonia, hydrogen sulphide and methane. In our case study, the influent and effluent of a membrane bioreactors (MBR) were examined, which belongs to the biological treatment stage. MBR is a membrane-filtration system that can be integrated directly into the aeration tank (Figure 3). Micro- and ultrafiltration membranes are used for this purpose [59] which separate the activated sludge, thus eliminating the need for the secondary clarifier.

The third purification stage is important to reduce different nutrients, such as phosphate and nitrogen, but especially for disinfection purposes. WWTP design often varies due to different boundary conditions, so that the elimination of phosphate in some WWTP belongs to the biological purification stage [60]. Various possibilities of third water treatment processes provide a final stage to meet specific requirements around the safe discharge of water. In our case, ozone was used as an oxidant. Although ozonation has proven to be an effective means of eliminating trace contaminants, undesired transformation products (TPs) are also formed in this process [28]. In this context, the formation of bromate is by far the most critical problem. The studied ozonation stage is controlled in a minimum operation with a volume proportional dosing of 2.0 mg O_3_/L. For the evaluation, samples collected before and after ozonation were compared over a period of two weeks.
Figure 3Overview flow scheme of the WWTP Aachen-Soers. Numbers represent the steps of the treatment process: (**1**) fine and coarse screen, (**2**) grit removal, (**3**) primary clarifier, (**4**) anaerobic denitrification, (**5**) aerobic nitrification, (**6**) secondary sedimentation, (**7**) ozonation, (**8**) second nitrification, (**9**) sand filter. (EF) effluent into surface water [61] (adapted with permission from [61]. Copyright 2022, Elsevier).
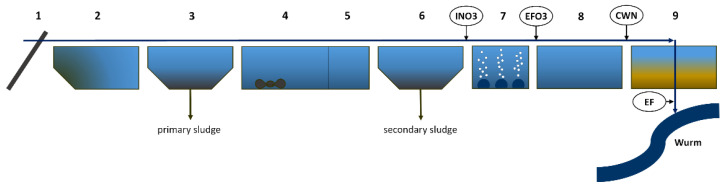



As described above, the most important removal pathways of organic compounds during wastewater treatment are biotransformation/biodegradation, abiotic removal by adsorption onto the sludge and stripping by aeration (volatilization) [62]. To track and review these interacting processes, sum parameter monitoring is utilized. Sum parameter bundling represents a summarized description of certain effects and substance parameters. Due to the large number of organic compounds in (waste)water, the substances are often not considered as individual substances but are grouped together according to their similar properties or effects [63]. This offers the advantages of easy, standardized and affordable measurements. Consequently, they belong to the most frequently measured parameters [27]. However, they do not provide any detailed information on the presence of toxic or emerging contaminants.

The BOD (biological or biochemical oxygen demand) indicates the amount of oxygen that is consumed by microorganisms for the biological, aerobic degradation of dissolved organic compounds in wastewater within a certain time. It is usually given in mg O_2_/L for a period of 5 days (BOD5) [64]. The source of BOD is readily degradable organic carbon and ammonia (NH_3_). These are common constituents or metabolic byproducts of plants and animal wastes and human activities [65]. The standard method for COD estimation is the so-called ‘Dichromate’ method [66,67] which is characterized by the acidification of the sample with sulphuric acid and the addition of silver sulphate. Due to the application of hazardous chemicals and an analysis time of 2 h, the method is not suitable for online use [68].

As a result of the implementation of the EU Industrial Emission Directive (IED), the total organic carbon (TOC) parameter is being introduced broadly into the monitoring of wastewater in various industries that reflects the organic load in a sample and is divided into dissolved organic carbon (DOC), undissolved organic carbon POC (particulate organic carbon), and volatile organic carbon (VOC) [69]. There is a slight preference of TOC, compared to the determination of COD, because TOC is subject to uniform regulation across Europe by the [70] standard, it is not hazardous, and it includes only organic carbon compounds. Generally, the determination of TOC is achieved by thermal or wet chemical oxidation so that CO_2_ is formed, which is subsequently measured by an NDIR (non-dispersive infrared) detector. The distinction between DOC and POC lies at a particle diameter of 0.45 μm [71]. On average, DOC represents 87% of TOC [72]. The DOC in wastewater is mainly composed of easily degradable substances, such as carbohydrates, proteins, fats and amino acids, as well as compounds that are difficult to degrade, such as cellulose and humic substances. Hence, particles which contain organic carbon should be considered during the measurement [70]. DOC content is reduced by up to 90% in WWTP plants. In Germany, however, there is no limit value for DOC in drinking water or wastewater, while Swiss drinking water may contain a maximum of 2 mg DOC/L [64].

Process analyzers, such as gas chromatographs with simple detectors [73,74,75], TOC-analyzers, spectroscopic probes and many others are very common in the process environment and have already been implemented successfully in (petro-)chemical plants or wastewater treatment plants for many years [76]. These techniques (e.g., GC, HPLC and IC systems) mainly use mass spectrometric detection which does not allow truly continuous monitoring. The spectral absorption coefficient (SAC254) provides information on the load of dissolved organic matter in the water. The coefficient is determined using a spectrophotometer at a wavelength of 254 nm and gives the absorption of the dissolved substances in the sample [77]. Modern optical sensors measure the turbidity according to EPA 180.1 [78] and ISO 7027 [79]. Other measuring probes apply UV-vis spectroscopy over the entire absorption spectrum from 190 to 750 nm. These sensors enable in-situ measurements during operation and real-time adjustments of the treatment process. However, the detection limit in waters with a complex composition and the difficulty of detecting the UV-Vis spectra of some pollutants in the water, such as suspended solids, dissolved inorganic substances and pathogenic microorganisms remain challenging [80]. Monitoring tools that interact continuously and automatically are not yet established. As [81] summarized: “Analytics 4.0 was proclaimed before it became a reality, and at least for the traditional analytical laboratory describes a vision rather than the actual status. Standardized protocols and commercial, unified solutions are required”.

Raman- and fluorescence laser spectrometry offers the potential to replace or supplement these methods and to establish faster and more efficient real-time monitoring as a new standard. This could act as an early warning system to detect targeted substances and allow rapid intervention.

## 3. Equipment

### 3.1. Raman/Fluorescence Detection System

For this study, we utilize a commercially available Photon Systems PL200 DUV spectrometer [47]. The system uses the benefits of deep-UV spectroscopy by using a NeCu transverse excited hollow-cathode gas laser, which leads to an interference-free spectral range without fluorescence and the auto-luminescence obscuration of Raman signals, and vice versa [45,47,48]. Most of the common devices that are used in spectroscopy are relatively large, heavy, and require significant cooling power. The PL 200 mitigates these limitations by utilizing a compact, low power, air-cooled hollow-cathode laser. The system is discussed in more detail in [43] and only the main components will be summarized. The laser emits light with a wavelength λ = 248.6 nm in 40 µs pulses with an average energy of 3.5 µJ. The device is designed to carry out fluorescence and Raman measurements which is enabled by the low wavelength, allowing a clear separation of the Raman shift signals and the autofluorescence of many natural materials.

Figure 4 shows a detailed illustration of the measurement setup. The incident laser light is focused by an array of mirrors where the aperture and light intensity are controlled by two iris diaphragms. An interference filter ensures a monochromatic light. Only one wavelength of 248.6 nm is transmitted through an additional built-in interference filter. The light that is reflected back from the sample reaches the laser-blocking lens via an achromatic objective. This lens almost reflects the Rayleigh radiation that is emitted by the sample and is transparent to the incoming Raman radiation. The sample is measured inside a quartz glass cuvette, a movable flow cell or placed onto a rotary table. The emitted fluorescence signal is then edge-filtered and spectrally separated by a monochromator. The light enters the monochromator through an entrance slit, the size of which can be adjusted. A larger slit decreases the spectral resolution. At the same time, enlarging the slit leads to an improved limit of detection (LOD). Since these two properties are linked ant proportionally, a trade-off exists between the resolution and LOD. We choose a relatively large slit size of 150 µm, in order to achieve better detection limits, since the LOD is much more important when trace amounts of substances need to be identified. After passing through the entrance slit, a 200 mm focal length Czerny–Turner spectrograph splits the light into the individual wavelengths. The fluorescence signal is acquired by selecting a 300 ln/mm grating, which yields a resolution of 1.2 nm. The 3600 ln/mm grating can be chosen, in order to investigate the sample’s Raman shift, which is investigated in a previous publication [43].

The resulting light spectrum is detected by a three-stage thermo-electrically cooled, back-thinned, back-illuminated CCD array with a resolution of 2048 × 128 pixels. The intensities are recorded with 16-Bit resolution, which means a maximum intensity of 65,000 counts can be recorded.

The emission is based on a back-scattering configuration (180° degree) when using the cuvette from Hellma^®^ or Starna Cell^TM^. For the detection of solid samples, such as microplastics or powders, a rotary table can be used, resulting in a 180s degree scattering configuration.

The spectrometer can be calibrated quickly and easily via software (Figure 5, left). The Raman grating (3600 ln/mm) uses Acetonitrile, a NIST-Standard (National Institute of Standards and Technology) for calibration. The calibration is based on two known peaks at 2250/3940 1/cm (compare Figure 5a) and can be verified with pure water (Photon Systems Inc. 2020). The fluorescence calibration is achieved by using a GaN-sample (gallium-nitride), showing one specific peak at 363 nm (Figure 5b). The calibration can be verified by measuring the amount of PHE (Phenylalanine) [82].

## 4. Data Processing

### 4.1. Data Preparation

All the data were processed and analyzed using the Spectrum Analyzer© software. The software is also used to control and calibrate the two holographic gratings. In addition, the software supports the use of CNN classifiers, which we trained using our measured spectra.

For the software optimization, each measurement is repeated 10 times, in order to improve the signal-to-noise ratio. The recorded spectra are processed to enable better comparability and to optimize the quality of the spectra. The fluorescence and Raman spectra are handled with the same mathematical algorithms. All spectra are smoothed using a Whittaker–Henderson algorithm. Additionally, sharp and intensive peaks, which can be caused by cosmic rays are removed (despiked). The intensities are then normalized regarding the pulse number and laser energy due to variations in the laser energy.

For the statistical analysis, each Raman or fluorescence measurement series, consisting of 10 individual measurements to evaluate the precision (and photodegradation or photobleaching) is preceded by a calibration process, as described before in Section 3.1. Dilution series with decreasing substance concentrations in the solution focus on the accuracy of the results in terms of trueness, and lead to the estimation of the limit of detection for each substance. Depending on the fluorescent properties, some of the substances are stronger diluted, meaning that they show more steps regarding their dilution (e.g., naproxen or tryptophan). The standard deviation (σ) and coefficient of variation (CV) describe the characteristics of the single spectrum and result in a correlation curve, if the results so indicate.
(1)$$\sigma =\sqrt{\frac{(\sum x-x^{\prime })2}{n}}$$
(2)$$CV=\frac{\sqrt{\frac{(\sum x-x^{\prime })2}{n}}}{x^{\prime }}$$
with *x* = intensity of signal, *x*′ = averaged intensity of signal, *n* = number of values.

The further processing refers to the fluorescence measurements, whereas Raman data processing was described in detail in a previous paper [43]. The averaged fluorescence spectrum of the individual concentration is used, from which the spectrum of the solvent has already been subtracted if the substance spectrum has a higher signal intensity than the solvent. In the other case, to prevent negative values for the resulting spectrum, we normalize the solvent spectrum to its initial, sharp peak (Figure 6). This procedure artificially increases the substance signal, in order to reduce masking by the solvent. If this procedure is necessary, a quantitative view on the date is no longer possible, regarding the limits of detection or the evaluation of concentrations.

In the further course, the maximum signal intensity of the different concentrated solutions is determined. The corresponding intensities are plotted against the concentration and a calibration curve is generated. From this, the mathematical detection limit of fluorescence spectroscopy for the respective substance can be determined graphically, as well as the accuracy of the measurements. For this purpose, the values of the calibration curve are used as reference values and the relative deviations of the measured spectra to this curve are calculated. If a linear regression results, the coefficient of determination (R^2^) can be used to estimate the accuracy. If the calibration curve does not yield a linear regression, the relative deviation is determined with Equation (3) [83].
(3)$$rel.\;\sigma =\frac{measured\;value-true\;value}{true\;value}$$

For the identification of the measured substance, the spectrum of the pure substance as a solid, gas or liquid is measured and stored by the database. References for Raman spectra can also be found in the literature or in open-source software [84]. A simple example for the fluorescence spectra of a dilution series of four concentrations of carbamazepine, diluted in ethanol, and measured in a quartz cuvette is shown in Figure 6.
Figure 6Fluorescence spectra of four carbamazepine concentrations, measured in a quartz cuvette and diluted in ethanol. The spectrum of the solvent ethanol (dotted line black), measured in a quartz cuvette and the empty cuvette itself (dotted line grey) are shown but not yet subtracted. Settings: pulse number 50; pulse frequency 40; slit size 150 µm; grating 300 ln/mm; focal length 20 mm (raw data from [85]).
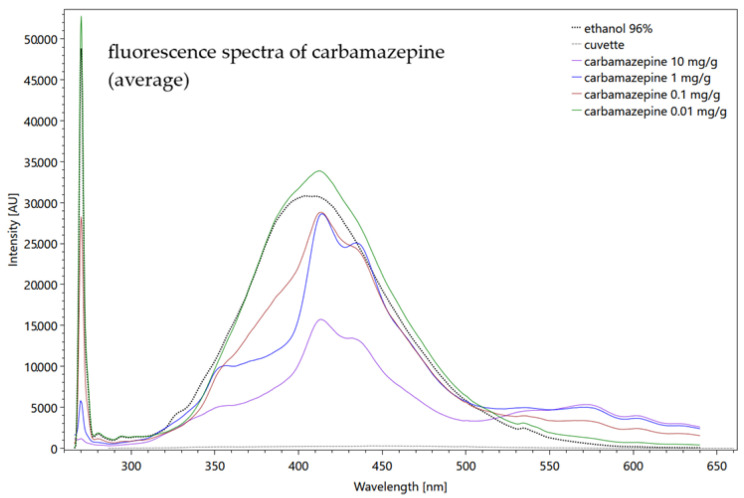



Figure 6 shows the dominance of the solvent ethanol and its influence on the carbamazepine signals. The highly concentrated 10 mg/g carbamazepine signal (purple line) is most attenuated by the solvent and the other substance spectra become increasingly similar to the ethanol spectrum as their concentration decreases (with the exception of 0.01 mg/g, green line). On the other hand, the signal of the quartz cuvette (dotted line grey) offers a very low intensity so that it can be neglected for further fluorescence measurements. The before-described procedure to reduce the influence of the solvent shows Figure 7, where the ethanol spectrum was subtracted and the averaged carbamazepine spectra were scaled and therefore artificially strengthened. As a result, the highly concentrated solution shows the highest intensity of more than 60,000 AU.

### 4.2. Substance Classification

In order to use the gathered spectra for monitoring purposes, the recorded substances must be identified. Various techniques for spectrum classification have been developed in the past few years. However, there is no wholly developed study applying these techniques to assist wastewater treatment operation [86]. The complex, nonlinear nature of spectroscopic data makes their analysis and classification challenging. A wide range of strategies for the analysis of spectral data exist, ranging from visual inspection by an expert through database matching, up to sophisticated machine learning models. In the analysis of complex mixtures, especially, the visual identification or correlational database-matching of spectra are often not applicable, since a superposition of all individual molecules’ fluorescence signal is recorded. Furthermore, manual classification is not a suitable workflow for the automated identification of substances. To reduce subjectivity and enable automated workflows we utilize a convolutional neural network (CNN) to classify the spectra. CNNs have proven to be suitable for the classification of spectral data and they need little data pre-processing, such as a baseline removal.

The architecture of our classifier is the same as the Raman spectra classifier that is introduced by [87] and which is also employed in our previous publication on Raman spectroscopy [43]. The only change in the network architecture is the number of output classes. Our investigations focused on a selection of 23 critical substances. The data pre-processing consists of resampling of the wavelengths to values ranging from 0 to 4000 and normalization of each individual spectrum’s intensities to a range between 0 and 1. A signal-to-noise ratio threshold of 3 was also implemented to avoid fitting the model to very noisy spectra. A total of 1700 spectra were used for training; 566 spectra were used as a test dataset. Convergence was achieved after 50 epochs of training using the Adam optimizer and categorical cross-entropy as the objective.

The resulting classifier has an accuracy of over 95% across all substances on unseen spectra. Performance across all classes was close to a 100% true positive rate, except for NaOH (sodium hydroxide) which is consistently misclassified, as well as tryptophan. Furthermore, the spectra of polymer suspensions can be subject to misclassification. This section may be divided by subheadings and it should provide a concise and precise description of the experimental results, their interpretation, and the experimental conclusions that can be drawn.

## 5. Methodology and Targets

The article at hand offers a methodical case study about a laser-induced Raman and fluorescence spectrometer (LIRFS) that is used in a wastewater treatment plant for monitoring purposes. The application in such a heterogenic and challenging environment is critical for an analytical device that needs enough sensitivity for detections limits down to a few µg/L, but also a robustness, so that the results are not affected by the presence of a variety of other chemicals and substances. The first step to test the suitability of the LIRF spectroscope was the Raman and fluorescence measurement of all the pure substances (as a powder or solid) and the substances in a solution varying in concentration, in order to evaluate reliable identification down to a concentration level, where a lower rate than 3/1 between signal and noise was observed. This was tested on different kinds of problematic micropollutants, such as pharmaceuticals, listed in the (Table 1) below, on two-amino acid tryptophan and tyrosine and on byproducts of the wastewater cleaning process (nitrate, nitrite, bromate, bromide).

In order to avoid the well-known problem of overlapping of the Raman spectrum by the generally more intense fluorescence spectrum, an excitation wavelength in the deep ultraviolet radiation range was selected. Fluorescence spectra that always emit at the same wavelength regardless of the excitation wavelength appear predominantly in wavelength ranges that are above 270 nm. In contrast, the Raman spectra show a dependence of the excitation wavenumber that is used. If the substrate is irradiated with deep ultraviolet light, the Raman spectra result in a fluorescence-free wavelength range that is below 270 nm [89]. Due to this fact, we could measure the Raman spectrum and the fluorescence spectrum of every substance in short succession only by changing the diffraction grating of the device, which takes a few seconds.

As mentioned before, we first measured the pure solids, e.g., powder, to obtain the typical substance spectrum for recognition in the database, and then set up a dilution series to evaluate the sensitivity of the spectrometer. Depending on the substance and its solubility, the solutions are produced with deionized pure water (Millipore^®^, purity ≥ 99%, Merck KGaA, Darmstadt, Germany), ethanol (C_2_H_6_O; purity ≥ 96%, Carl Roth GmbH & Co. KG, Karlsruhe, Germany) or as standardized solutions for ion chromatography. Therefore, the optical properties of the solvent and the used cuvette (quartz glass) must be considered, or their signal must be subtracted from the resulting spectrum. Pre-tests showed a light optical effect of the quartz cuvette for the fluorescence measurements (maximal 600 AU), whereas there is one regarding the Raman measurements (Figure 8). In order to bypass this obstacle, the heterogeneous WWTP samples regarding the byproducts were measured in a flow cell that was constructed with sapphire glass, which had no effect on either fluorescence or Raman scattering.

Regarding WWTP sampling and sample preparation, all the wastewater samples were collected over a maximum of 24 h and mixed without any other treatment but keeping them cool and dark in the refrigerator. Figure 9 shows the extraction points for the different targets coming from two different WWTP.

Regarding online wastewater monitoring, another issue is the evaluation of the assumed correlation between the sum parameter DOC and the fluorescence intensity in the biological treatment stage [91,92,93]. The evaluation of these in a literature-published hypothesis revealed a correlation between the tryptophan-like peak and DOC, which was another focus of the current investigation. According to [94], tryptophan is especially useful as a reporter for conformation changes in proteins, since several protein changes can result in predictable changes in tryptophan fluorescence in a sample [94]. On the other hand, [95] found strong correlations between tryptophan and the colony counts of a variety of indicator species, including E. coli. bacteria, which are extremely important for wastewater disinfection, especially since online monitoring is not yet state of the art.

## 6. Results

The results of this study originate from different field and laboratory campaigns. All the samplings and experiments were performed under standardized conditions so that comparability could be assumed. Deviating framework conditions are explicitly mentioned and are justified in the respective objective of the experiment. The selected results underline the possibility to apply the evaluated spectrometer for online monitoring purposes, but they also emphasis difficulties and obstacles in operation.

We first examined the capability of fluorescence intensity measurements to correlate with organic carbon concentrations. According to [93], fluorescence peaks C in water can be used to estimate DOC, where peak C correlates with the content of dissolved organic carbon in the water. Our results could confirm these observations and transfer them to treated wastewater (Figure 10). The fluorescence analysis of wastewater samples from two different WWTPs after passage through the membrane bioreactor are shown in Figure 9 [96]. The marked spectra correspond to the DOC concentrations that are estimated in the collaborating laboratory via thermal oxidation and NIR detection DIN EN 1484. There is a clear tendency that the higher the DOC concentration, the higher the peak intensity (except for sample 163). The coefficient of variation from these seven samples ranges from 0.5 to 4.7.

In our measurements, the coefficient of determination R^2^ is 0.85 (Figure 11), referring to seven effluent samples of two different WWTPs that are equipped with a membrane bioreactor. This corresponds to the results from the literature, e.g., from [97] of R^2^ = 0.8 and 0.81 from [98]. The correlation between the fluorescence intensity in all our measurements and the DOC concentration determined in the laboratory showed that a differentiated consideration of the extraction point was useful. The correlation coefficient of all the samples throughout the purification processes was 0.74 [96]. Variations show up more clearly in lower concentration ranges than in higher ones, but the results suggest that the correlations allow a rough estimation of the concentrations of DOC, and every determination of DOC in the lab could be reproduced by the spectrometer. In particular, anomalies—such as a sudden increase in DOC—can be confidentially detected that allow early warning and a rapid response, for example, the application of ozone.

With the aid of the dilution series, the detection limit for the amino acids tyrosine and tryptophan could be determined. Tyrosine proves to be detectable up to 0.05 mg/L, but for tryptophan the spectrometer shows a much greater sensitivity down to 0.001 mg/L (Figure 12b). At lower concentrations, the intensities continue to decrease, and the peaks are strongly affected by solvent and background noise.

For comparison the results of tyrosine is shown in Figure 13.

No correlation was found between the tyrosine-like peak and DOC in wastewater, which is probably because tyrosine fluoresces at an excitation wavelength that is lower than 248.6 nm and the presence of quenchers in the wastewater reduces the fluorescence properties of tyrosine. Regarding the correlation between the tryptophan-like peak in the fluorescence measurements and DOC, ref. [99] showed a high correlation between the (tryptophan-like) peak and DOC concentration (R^2^ > 0.85) and concluded that this peak can be used for monitoring DOC in untreated and treated water. Our results relativize this observation, depending on our excitation wavelength, and suggest a more differentiated approach regarding treated wastewater. Overall, a medium correlation can be demonstrated between the DOC concentration and the peak intensities of T2 (Figure 14).

Considering all samples, the coefficient of determination is R^2^ = 0.75. This result is largely in line with the literature values [94,95]. Differentiating the extraction points shows that the correlation is significantly higher for the effluent samples (R^2^ = 0.95) than for the influent samples (R^2^ = 0.19). This leads to the conclusion that the correlation is higher for water with a low DOC concentration (≤8.3 mg/L in our samples) than for samples with a higher DOC content (≥37 mg/L in our samples). This may also be related to the presence of other interfering substances in the water which are removed in the aeration tank, but strengthens the previous results of the membrane bioreactor.

Analyzing the spectra of heterogeneous WWTP samples is extremely challenging due to effects that are caused by coexisting substances and their interactions. Correlations do not appear to be consistently linear, and the spectra interact with substances that remain in the treated wastewater. Figure 15 shows six influent and six effluent samples from the biological treatment stage. Samples with a similar color were taken on the same day (influent samples have a solid line; effluent samples’ lines are dotted) at the entry and at the output of the aeration tank.

The reduction rates of tryptophan peak T2 through purification vary for the sample pairs from a maximum of 80.0% (sample number 634 and 635) to a minimum of 32.2% (sample number 706 and 707) (Figure 15). Effluent samples number 703 and 707 are characterized by a low purification efficiency and lead to high contents of higher-ring hydrocarbons in the wavelength of DOC with an emitting wavelength of around 440 nm. The corresponding reduction in the DOC rates shows a similar trend, but the amounts are not transferable. The DOC reduction in sample pair number 634 and 635 amounts to 90.6%, and for sample number 706 and 707 the DOC content was reduced by 79.3%, which differs from the decrease in tryptophan content in the same sample pair. The variation coefficient for the calculated DOC concentrations referring to the correlation curve in Figure 14 leads to values of more than 200% in some cases.

Regarding the pharmaceutical micropollutants, we demonstrate relevant results showing the ability of the spectrometer to detect specific substances, and its suitability for monitoring purposes. The spectra of metformin hydrochloride as a solid shows the trend of photodegradation after 10 measurements (Figure 16).

The metformin hydrochloride dilution series do not show very meaningful results in the fluorescence and Raman measurements, but low signals down to 0.01 g/L are detectable. The metformin hydrochloride signals are strongly affected by the water emissions, but for the Raman measurements a clear peak at 940 1/cm was recognized that was not superimposed by the Millipore^®^ spectrum. Carbamazepine by far shows stronger fluorescence properties. Upon measuring the solid powder, the CCD detector is completely oversaturated so that the full spectrum is not visible. Dilution was carried out with 96% ethanol. Figure 17 shows the fluorescence and Raman spectra of the diluted carbamazepine.

Remarkable in both measurements is the influence of ethanol on the carbamazepine spectra in such a way that the peaks of ethanol drag along the spectra of carbamazepine, especially in the low concentrations. Acetaminophen shows an intensive Raman peak at 1607 1/cm, measured with 2000 pulses. After the correction of the rolling baseline, the variation coefficient remains relatively high (12.8%) due to slight photodegradation effects during 10 repetitions. The fluorescence measurements with 100 pulses could confirm high variations but without a clear trend. The fluorescence dilution series showed good differentiation between the concentration steps and were detectable down to 0.1 mg/g.

Hydrochlorothiazide powder is specifically recognizable by its fluorescence signal with a strong peak at 373 nm (measured with five pulses) with an intensity of more than 45,000 AU and a good precision that is represented by a variation coefficient of 1.9%. However, the dilution series revealed no correlation between the concentration and signal intensity, so that the construction of the calibration curve was not useful.

On the contrary, naproxen is detectable in solution with ethanol in the fluorescence spectrum down to 1 × 10^−4^ mg/g (Figure 18). Theoretically, this enables its application in the WWTP monitoring. Even the Raman measurements show that two characteristic peaks at 1373 1/cm and 1616 1/cm recognize naproxen specifically.

Regarding the Raman measurements (Figure 19) for the nitrate dilution series, a pulse number of 500 was used for the solutions containing 50 mg/L, 25 mg/L, 10 mg/L and 5 mg/L NO_3_^−^. On the other hand, for the solution with 1 mg of NO_3_/L, a pulse number of 1000 was used. The corresponding spectrum was then scaled by a factor of 0.5 to establish comparability with the other spectra in the dilution series. This procedure is generally acceptable because there is a linear relationship between the applied pulse number and the measured intensity. However, problems can occur in the range of high concentrations due to nonlinear behavior in the form of strong absorption and reemitting.

The applied function of the calibration curve (Figure 20) is concave, shows a very good correlation with the measured values with a coefficient of determination of 0.99 and can predict the measured peak intensities accordingly well. The relative deviation of the data around the arithmetic mean ranges from 2.31% to 3.97% with an average variation coefficient of 3.03%, indicating a high level of precision.

Nitrate is not detectable in the fluorescence measurements but is clearly and specifically detectable down to 10 mg/L of concentration (Figure 19b, green line) in the Raman measurements. This is not convincing for the WWTP application for the effluent water, even though values of this concentration sometimes exist. Further, of course, an anomaly warning in case of unexpected loads entering the wastewater treatment plant is possible as pre-studies from the food industry have indicated. The same behavior of nitrate is found for nitrite showing a lacking fluorescence emission; even the Raman intensities are little lower than for the nitrate spectra (Figure 21).

Regarding the Raman dilution series of nitrite, a reduction in the peak intensity is shown with each dilution step, without exception. Up to a nitrite concentration of 10 mg/L a clear peak is formed in the spectra. At a nitrite concentration of 5 mg/L only a broadened base of higher intensity can be seen. The applied function of the calibration curve is convex, shows a very good correlation with the measured values with a coefficient of determination of 0.98 and can predict the measured peak intensities accordingly well with an average variation coefficient from 2.27.

A more problematic substance to detect is bromide, which was also measured in the flow cell. Since the measured spectra of the dilution series do not show any peak that could originate from the bromide. Furthermore, the measured spectra differ only marginally from each other, despite strongly different bromide concentrations. The expected Raman wavenumber range of the bromide anion agrees with the statement of [100] that most anions of inorganic salts have peaks in the wavenumber range of 500–1100 1/cm. Since these peaks occur in the region of the enveloping curve of the water, there is a possibility that the bromide peak is superimposed by the intense signal of the water. Another possible explanation is provided by the basic operation of Raman spectroscopy, since the Raman signals measured are primarily due to vibrations of the electron bonds between two atoms of a molecule. However, in the bromide standard that was used, the potassium bromide (KBr) was dissolved in water and therefore exists in its dissociated form as Br^-^ anions and K^+^ cations. Consequently, no electron bond exists between the bromide and another atom that could be excited to vibrate.

The bromate dilution series was carried out analogously (Figure 22) to the bromide dilution series. Here, too, a total of six solutions with different concentrations were prepared and sampled as standard with a pulse number of 500. The two solutions with the lowest concentrations were analyzed with a pulse number of 1000 for visibility reasons.

The spectra of the bromate dilution series show similarity to the bromide series in that there are no specific peaks in the spectrum that cannot be found in the spectrum of the solely Millipore^®^ water. Figure 23 shows the reference spectrum of solid potassium bromate (KBrO_3_). The spectrum shows that no Raman-active signals exist above a wavenumber of 900 1/cm. The peak with the highest intensity occurs at a wave number of approx. 800 1/cm.

The Raman signals occurring in the wavenumber range between 150 1/cm and 200 1/cm are attributed by [101] to the vibration of the crystal lattice of the solids and therefore do not occur when the bromate is presented in its dissociated form. The two most intense Raman peaks of the bromate anion represent the ν1 and ν4 vibrations, which occur at wavenumbers of about 800 1/cm and 420 1/cm, respectively. Thus, it can be stated that the bromate anion, in contrast to the bromide anion has Raman-active vibrations and can in principle be detected with the applied technique, but the peaks of the vibrations are not visible in the measured spectra. The most likely explanation is that the peaks of the bromate anion are superimposed by the enveloping curve of the water.

Summarizing the results of all measurements, Table 2 gives an overview of the performed analyses and the ability of the spectrometer that was used to detect the different substances.

## 7. Discussion and Conclusions

The presented results are manifold. It should be noted that both measuring techniques that are included in one device have specific strengths and weaknesses. The fluorescence technique is very suitable for observing DOC changes in wastewater. This technique can provide a good service to wastewater treatment plants by offering online monitoring that checks the proper operation of the purification process and sending a warning when it is threatened by sudden heavy pollutant loads. This could be proven by the investigation results in the aeration tank, the membrane bioreactor and at the ozonation. The results prove that changes in concentration ranges of a few mg/L DOC could be observed, and these correspond to the results of the cooperating laboratory. Generally, it could be confirmed that high concentrations of various co-existing substances, as in wastewater, lead to the underestimation of substance quantities (by fluorescence), probably due to the inner filter-effect and quenching. Another weak point becomes apparent in the identification of individual substances. For example, fluorescence can detect changes very sensitively, and it can indicate which substance transformations have taken place, but this does not lead to the proper identification of substances. Here, only the observation of substance groups can be used to determine these substances based on their typical wavelengths.

The Raman measurements impressively demonstrated that selected pharmaceutical substances (carbamazepine, naproxen, tryptophan) could be detected down to a few µg/L range—this is a concentration range that is within the range of real application in a sewage treatment plant. However, some of these micropollutants indicated that due to their low Raman activity, or even superposition by the water/solvent signal, the detection limits were possible only up to the mg/L range or even the g/L (metformin hydrochloride) range, making it impossible for use in wastewater monitoring. Furthermore, the monoatomic anions (e.g., bromide) of dissolved salts are a problem in Raman spectroscopy since they do not provide a direct Raman signal. Thus, it must be concluded that the monitoring of these substances using Raman spectroscopy is not practical. In the instance of nitrate, the detection limit seemed somewhat weak (roundabout 10 mg/L); however, the Raman detection of nitrate after biological purification in a wastewater sample was successful, as [43] shown. This exact identification is possible due to the AI support which could reach a successful assignment rate of over 95%. Once a substance is trained by the CNN and stored in the database, its identification becomes much easier and more reasonable. For this reason, LIRF spectroscopy seems to be a promising tool for online monitoring, and even device-specific adjustments like the variation of the excitation wavelength could be particularly helpful for the application case of a wastewater treatment plant.

## Figures and Tables

**Figure 1 sensors-22-04668-f001:**
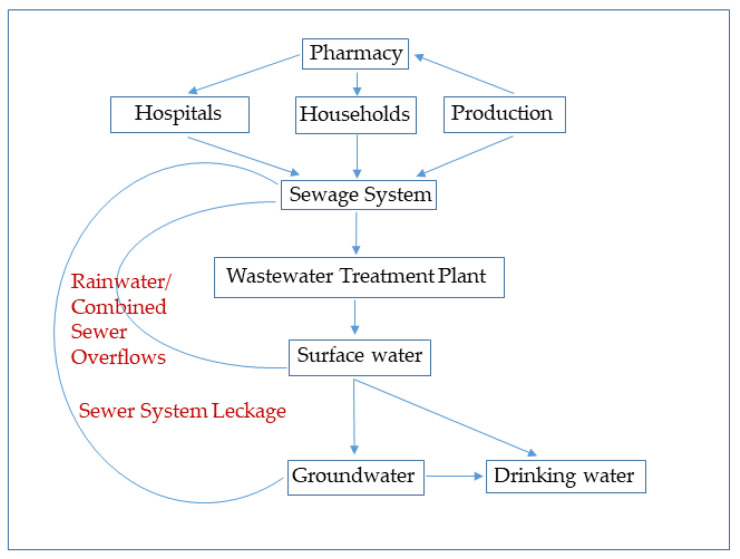
Material flow analysis of micropollutants in the environment (adapted with permission from [19]. Copyright 2010, IKSR—CIPR—ICBR—all rights reserved).

**Figure 2 sensors-22-04668-f002:**
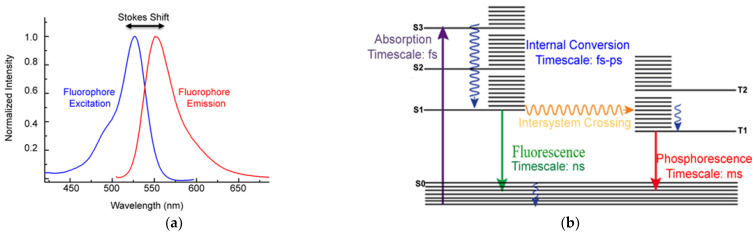
Wavelength shift due to Stokes shift, (**a**); Jablonski diagram, (**b**) (adapted with permission from [40]. Copyright 2015, American Chemical Society).

**Figure 4 sensors-22-04668-f004:**
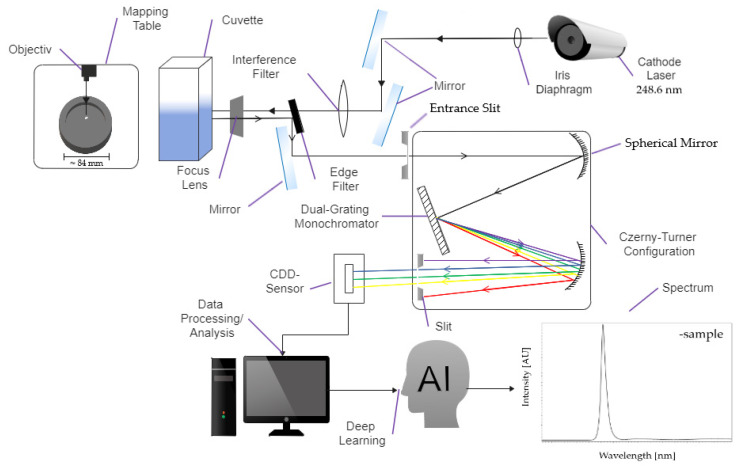
Optical setup of the Deep-UV Raman and Photoluminescence 200 Spectrometer visualized with Wondershare EdrawMax Copyright © 2022 Edrawsoft.

**Figure 5 sensors-22-04668-f005:**
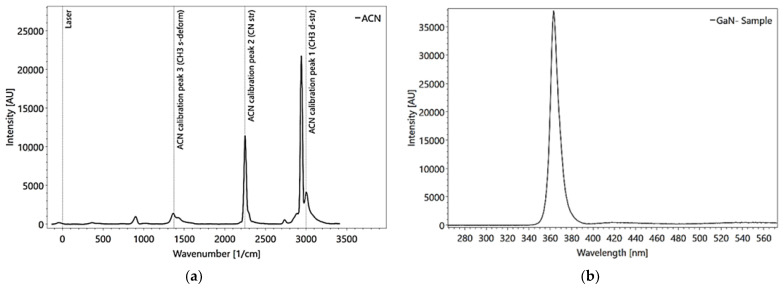
Raman calibration using Acetonitrile (Spectrum Analyzer) measured with 30 pulses (**a**). Fluorescence calibration using a GaN-sample (Spectrum Analyzer), measured with 1 pulse, (**b**).

**Figure 7 sensors-22-04668-f007:**
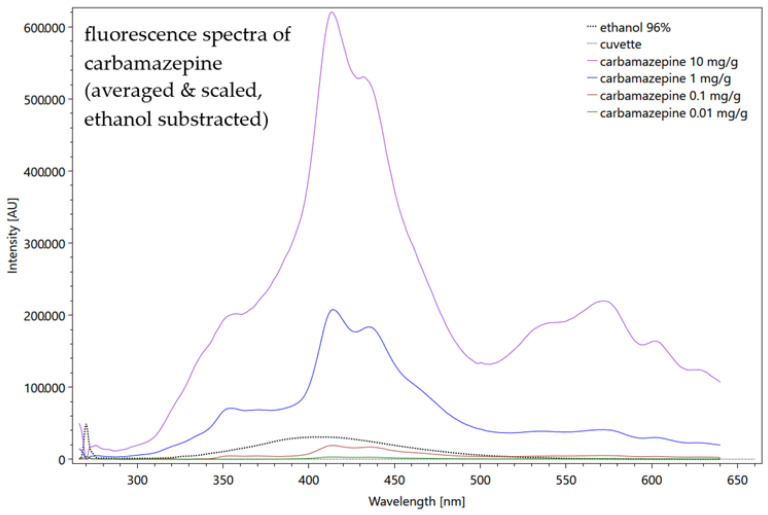
Fluorescence spectra of four carbamazepine concentrations, averaged from 10 repeated measurements. The spectra were scaled and the ethanol spectrum was subtracted. The spectrum of the solvent ethanol (dotted line black), measured in a quartz cuvette and the empty cuvette itself (dotted line grey) are shown. Settings: pulse number 50; pulse frequency 40; slit size 150 µm; grating 300 ln/mm; focal length 20 mm (raw data from [85]).

**Figure 8 sensors-22-04668-f008:**
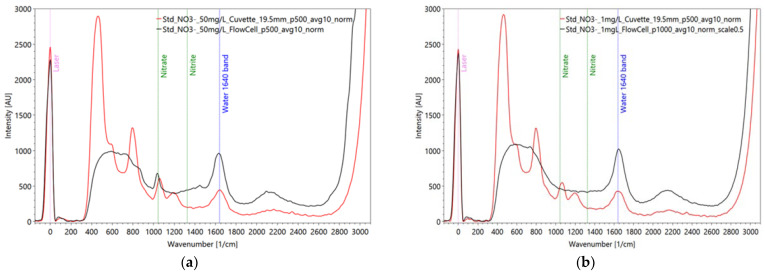
Influence of the measurement set-up on the signal: Raman spectrum of two nitrate solutions with 50 mg/L (**a**) and 1 mg/L NO_3_^-^ (**b**) in a cuvette (red line) and in a flow cell (black line). Critical interpretation arises at the nitrate reference peak at around 1050 1/cm because cuvette and nitrate signal appear at the same wavenumber, indicating the presence of higher nitrate concentrations. Settings: pulse number 500; pulse frequency 40/s; slit size 150 µm; grating 3600 ln/mm; focal length 20 mm (only cuvette) (raw data from [90]).

**Figure 9 sensors-22-04668-f009:**
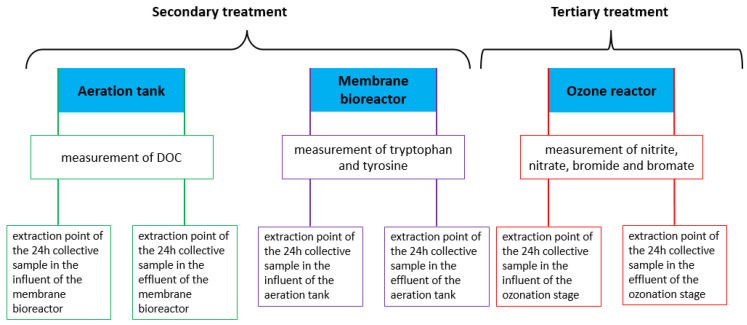
Overview of the extraction points of the 24-h collective samples of two different wastewater treatment plants and their targets around the biological treatment (aeration tank and membrane bioreactor) and the ozonation.

**Figure 10 sensors-22-04668-f010:**
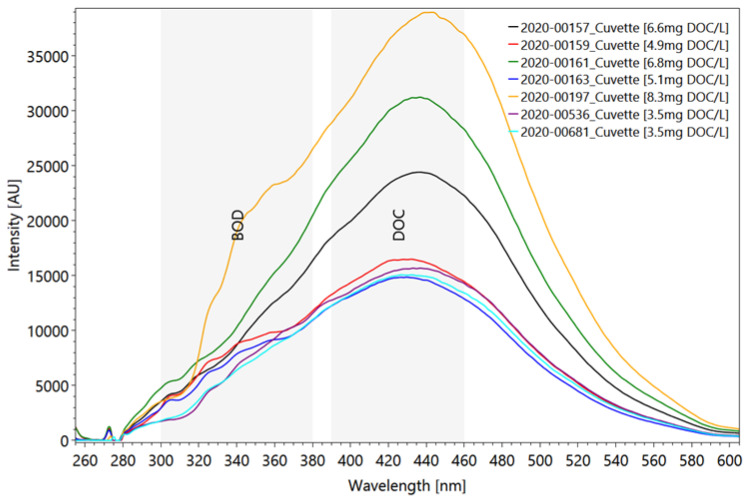
Averaged and scaled fluorescence analysis of 7 wastewater samples after the treatment in a membrane bioreactor [96] and as reference, DOC concentrations of the same samples in [mg/L] analyzed by a collaborating laboratory. Settings: pulse number 20; pulse frequency 40/s; slit size 150 µm; grating 300 ln/mm; focal length 20 mm, measured in a quartz glass cuvette (raw data from [96]).

**Figure 11 sensors-22-04668-f011:**
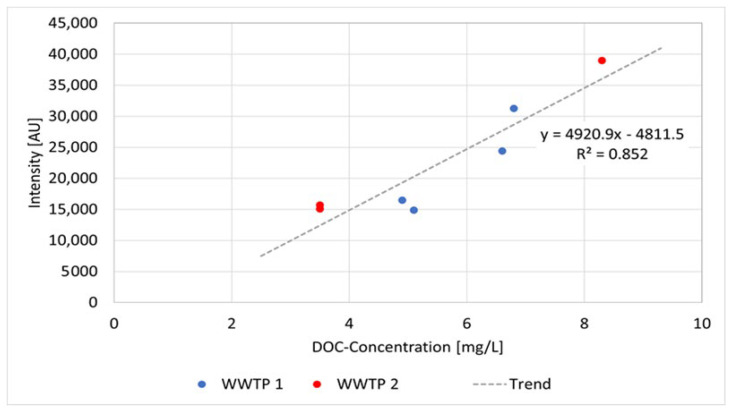
Correlation curve, functional equation, and coefficient of determination between DOC and fluorescence peak intensity from 7 effluent samples exiting the membranes bioreactor from two different WWTPs (raw data from [96]).

**Figure 12 sensors-22-04668-f012:**
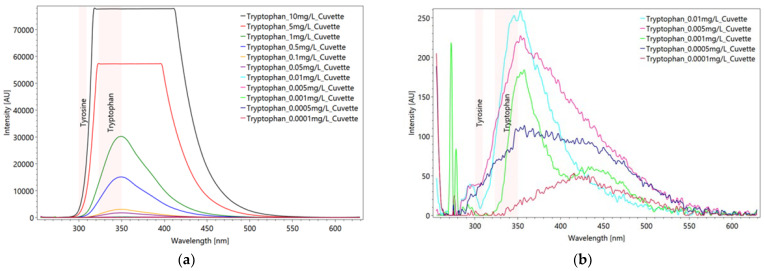
Dilution series from averaged and scaled tryptophan spectra (**a**) after subtraction of solvent spectrum. Due to oversaturation of the CCD detector the 10 mg/L and 5 mg/L concentrated solution spectra are cut off. In the *Y*-axis enlargement (**b**) the detection limit of tryptophan is clearly seen at the signal of 0.001 mg/L (green line). Settings: pulse number 10, pulse frequency 40/s, focal length 20 mm, measured in a quartz glass cuvette (raw data from [63]).

**Figure 13 sensors-22-04668-f013:**
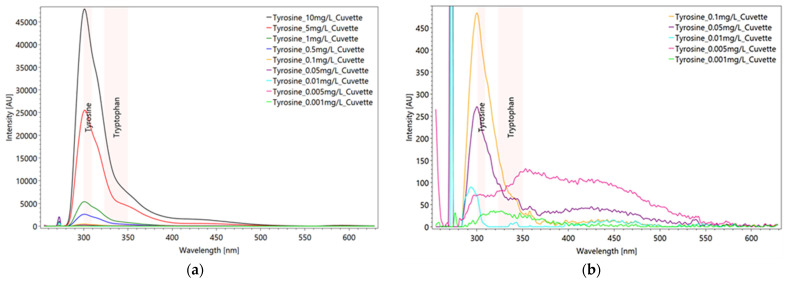
Dilution series from averaged and scaled tyrosine spectra (**a**) after subtraction of solvent spectrum. Enlargement of the *Y*-axis emphasizes the detection limit of the substance (**b**). Settings: pulse number 10, pulse frequency 40/s, focal length 20 mm, measured in a quartz glass cuvette (raw data from [63]).

**Figure 14 sensors-22-04668-f014:**
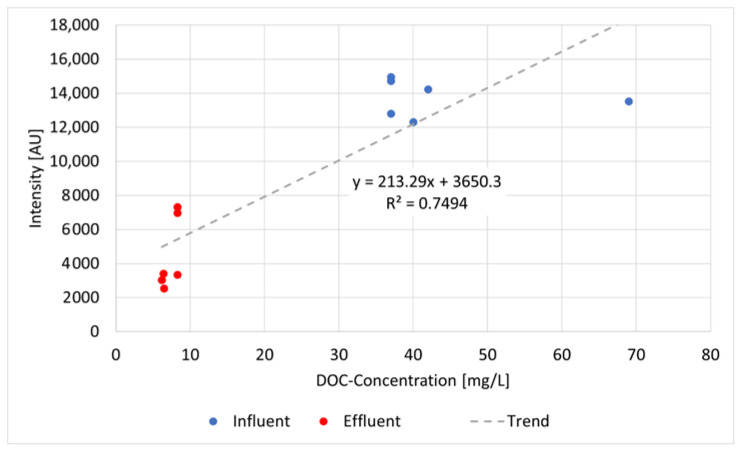
Correlation curve, functional equation, and coefficient of determination between DOC and the intensities of fluorescence spectra of peak T2 of all influent and effluent samples around the aeration tank (raw data from [63]).

**Figure 15 sensors-22-04668-f015:**
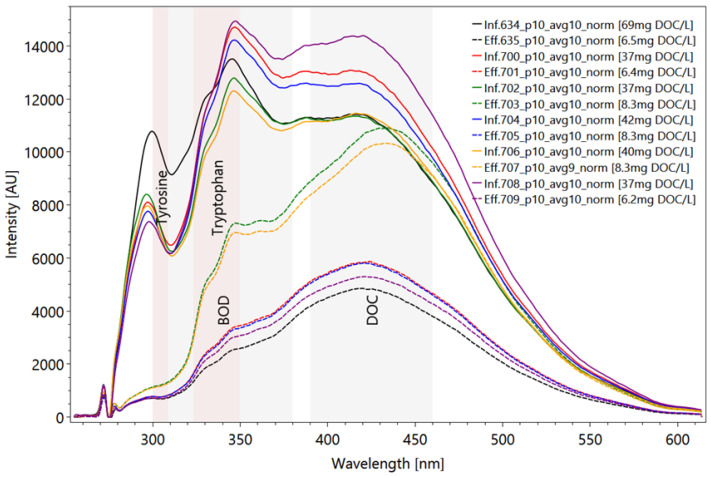
Comparison between the fluorescence spectra of WWTP influent (inf.) (solid lines) and effluent (eff.) (dotted lines) samples of the same sampling day; measurement results after complete processing including subtract spectrum. The reddish background marks the characteristic tyrosine and tryptophan peak areas; the grey background represents the characteristic peaks of DOC and BOD. Influent and effluent samples with sample number and content of DOC (in square brackets), determined in a collaborating laboratory. Settings: pulse number 10; slit size 150 µm; grating 300 ln/mm; focal length 20 mm; pulse frequency 40/s (raw data from [63]).

**Figure 16 sensors-22-04668-f016:**
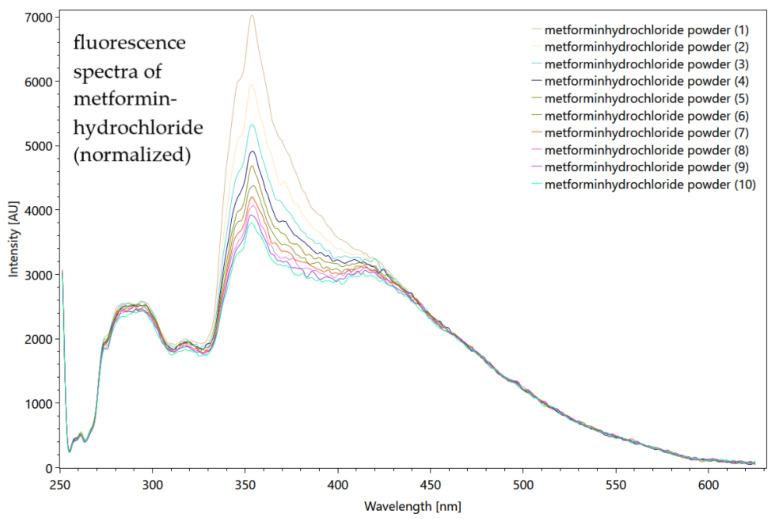
A total of 10 repeated measurements of fluorescence of metformin hydrochloride as a powder, showing slight photodegradation phenomenon. Settings: pulses number 200, grating 300 ln/mm, slit seize 150 µm, pulse frequency 40/s, measured on a turning table (raw data from [85]).

**Figure 17 sensors-22-04668-f017:**
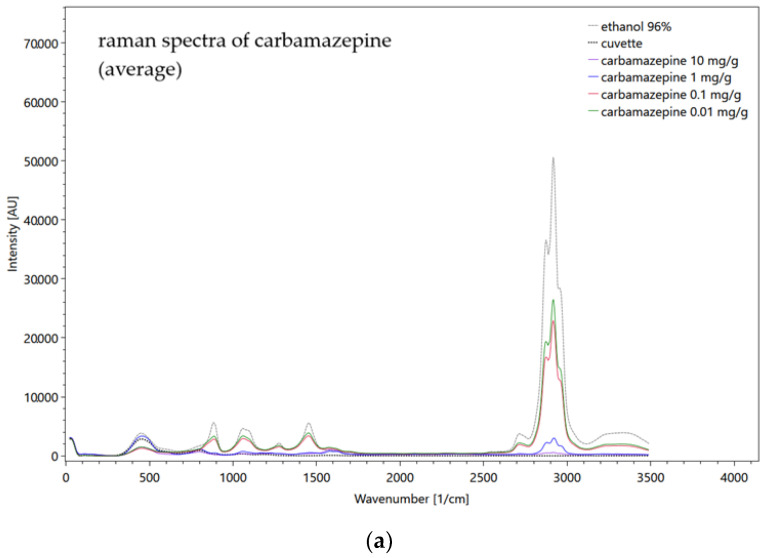
Averaged Raman (**a**) and fluorescence (**b**) spectra of carbamazepine before subtracting ethanol spectrum and scaling, including the specific spectrum of the empty cuvette and the solvent ethanol in the same cuvette. Settings above: pulse number 500, slit seize 150 µm, grating 3600 ln/mm, focal length 20 mm (in cuvette); settings bottom: pulse number 50, slit seize 150 µm, grating 300, focal length 20 mm (in cuvette) (raw data from [85]).

**Figure 18 sensors-22-04668-f018:**
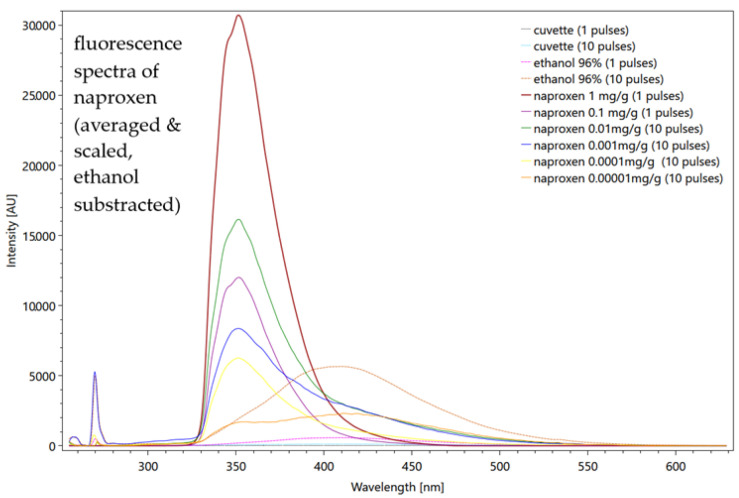
Averaged, scaled fluorescence spectra of naproxen after subtraction of ethanol. The measurements also emphasize the influence of the number of pulses for cuvette and ethanol. Settings: pulse number 1/10, slit seize 15 µm, grating 300, focal length 20 mm (raw data from [85]).

**Figure 19 sensors-22-04668-f019:**
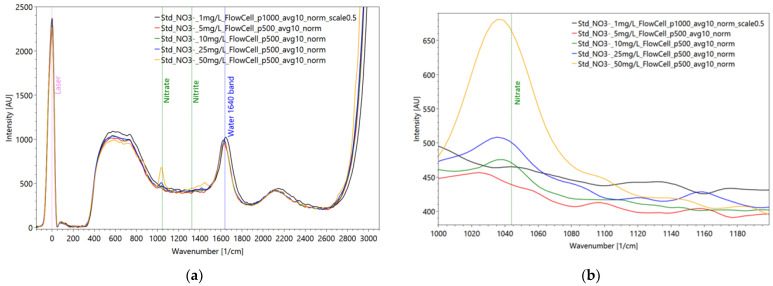
Raman measurements of dilution series of averaged nitrate spectra, measured in a flow cell to avoid cuvette disturbance (**a**) and a detailed enlargement of the nitrate reference peak at 1050 1/cm (**b**); settings: pulse number 500/1000, slit seize 150 µm, grating 3600 ln/mm, pulse frequency 40/s (raw data from [90]).

**Figure 20 sensors-22-04668-f020:**
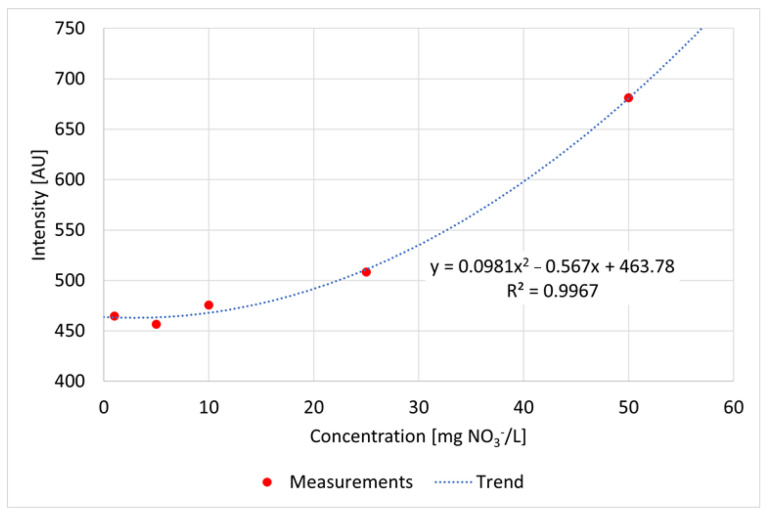
Correlation curve, functional equation and coefficient of determination between nitrate concentrations and Raman peak intensities referring to the dilution series of nitrate (raw data from [90]).

**Figure 21 sensors-22-04668-f021:**
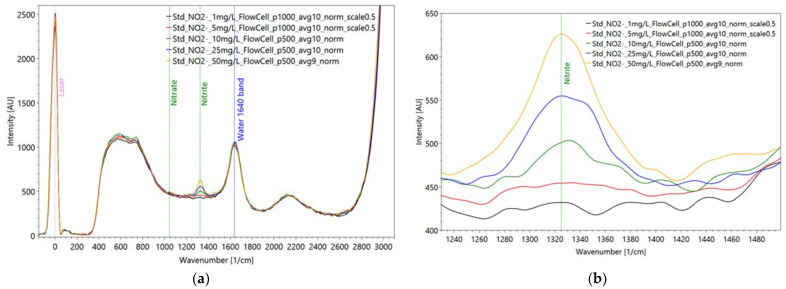
Raman measurements of dilutions series of averaged nitrite spectra, measured in a flow cell to avoid cuvette disturbance (**a**) and a detailed enlargement of the nitrite reference peak at 1325 1/cm (**b**); settings: pulse number 500/1000, slit seize 150 µm, grating 3600 ln/mm, pulse frequency 40/s (raw data from [90]).

**Figure 22 sensors-22-04668-f022:**
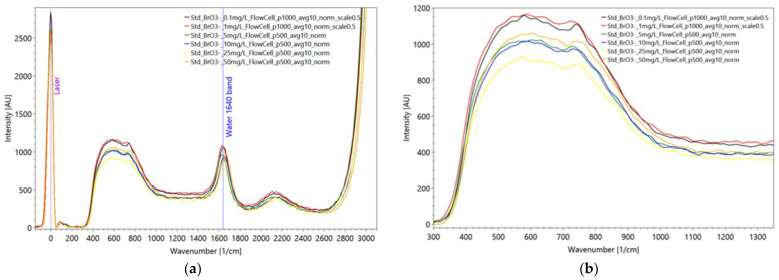
Averaged Raman bromate spectra of the dilution series with 6 different concentrations (**a**) and detailed enlargement of the enveloping curve of the water signal from 400 to 800 1/cm and its low effect of different bromate concentrations (**b**). Settings: pulse number 500/1000, slit seize 150 µm, grating 3600, pulse frequency 40/s (raw data from [90]).

**Figure 23 sensors-22-04668-f023:**
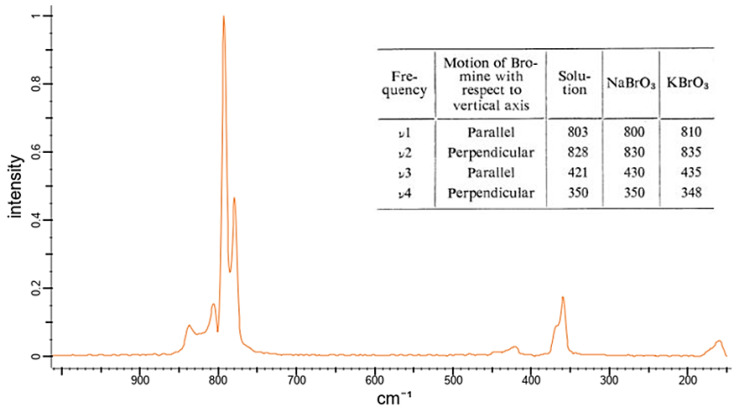
Raman reference spectrum of solid potassium bromate (KBrO3) from the spectrabase.com database and table with Raman active signals of solid bromate compounds and dissolved bromate anions, according to [101]. (Adapted with permission from [101]. Copyright 1973, Journal of the Physical Society of Japan (JPSJ)).

**Table 1 sensors-22-04668-t001:** Measured substances with manufacturer information, concentrations of the dilution series and given limits.

Substance; CAS no.	Manufacturer	Measurement	Concentration	Legal Limits
**Metformin hydrochloride**; 1115-70-4; purity > 99%	BioTrend, Cologne, Germany	Pure powder in solution with cuvette from Starna Cells^®^ Atascadero, CA, USA	Pure substance and 10; 1; 0.1; 0.01 g/L in solution with Millipore^®^, Merck KGaA Darmstadt, Germany	Preventive value 0.1 µg/L °
**Carbamazepine**; **297-46-4**; purity > 98%	BioTrend, Cologne, Germany	Pure powder in solution with cuvette from Starna Cells^®^ Atascadero, CA, USA	Pure substance and 10; 1; 0.1; 0.01 mg/g in solution with ethanol	Preventive value 0.1 µg/L °
**Hydrochlorothiazide**; 58-93-5; purity > 99%	BioTrend, Cologne, Germany	Pure powder in solution with cuvette from Starna Cells^®^ Atascadero, CA, USA	Pure substance and 10; 1; 0.1; 0.01 mg/g in solution with ethanol	Preventive value 0.1 µg/L °
**Acetaminophen**; 103-90-2; purity > 99%	BioTrend, Cologne, Germany	Pure powder in solution with cuvette from Starna Cells^®^ Atascadero, CA, USA	Pure substance and 10; 1; 0.1; 0.01 mg/g in solution with ethanol	Preventive value 0.1 µg/L °
**Naproxen**; 22204-53-1; purity > 98%	BioTrend, Cologne, Germany	Pure powder in solution with cuvette from Starna Cells^®^ Atascadero, CA, USA	Pure substance and 0.005; 0.001; 0.0005; 0.00001 down to 1 × 10^−5^ mg/g in solution with ethanol	Preventive value 0.1 µg/L °
**Diclofenac**; 15307-93-4; purity > 98%	BioTrend, Cologne, Germany	Pure powder in solution with cuvette from Starna Cells^®^ Atascadero, CA, USA	Pure substance and 10; 5; 1; 0.5; 0.1; 0.05; 0.01 g/L in solution with Millipore^®^, Merck KGaA, Darmstadt, Germany	Water hazard class 3 ^+^ acc. to WDF watch list ^X^
**Bromate**; 7789-38-0	VWR Chemicals	In solution with solvent water, in flow cell	50; 25; 10; 5; 1; 0.1; 0.01 mg/L in solution with Millipore^®^, Merck KGaA, Darmstadt, Germany	10 µg/L, according to Directive EU #
**Bromide**; 7647-15-6	Supelco^®^, Merck KGaA, Darmstadt, Germany	In solution with solvent water, in flow cell	50; 25; 10; 5; 1; 0.1; 0.01 mg/L in solution with Millipore^®^ Merck KGaA, Darmstadt, Germany	-
**Nitrate**; 7631-99-4	ROTI^®^Star Carl Roth GmbH + Co. KG, Karlsruhe, Germany	In solution with solvent water, in flow cell	50; 25; 10; 5; 1 mg/L in solution with Millipore^®^, Merck KGaA, Darmstadt, Germany	50 mg/L, according to WFD ^X^ and Directive EU #
**Nitrite**; 7632-00-0	Certipur^®^ Merck KGaA, Darmstadt, Germany	In solution with solvent water, in flow cell	50; 25; 10; 5; 1 mg/L in solution with Millipore^®^, Merck KGaA, Darmstadt, Germany	0.5 mg/L, according to Directive EU #
**Tryptophan**; 73-22-3	Merck KGaA, Darmstadt, Germany	In solution and as a solid, cuvette from Starna Cells^®^	Pure substance and 10; 5; 1; 0.5; 0.1; 0.05 0.01; 0.005; 0.001 mg/L in solution with Millipore^®^, Merck KGaA, Darmstadt, Germany	-
**Tyrosine**; 60-18-4	Carl Roth^®^ Carl Roth GmbH + Co. KG, Karlsruhe, Germany	In solution and as a solid, cuvette from Starna Cells^®^	Pure substance and 10; 5; 1; 0.5; 0.1; 0.05 0.01; 0.005; 0.001 mg/L in solution with Millipore^®^, Merck KGaA, Darmstadt, Germany	-

° Preventive value, according to LANUV 2015 and https://www.flussgebiete.nrw.de/monitoringleitfaden-oberflaechengewaesser-anhang-d4-7724 (accessed on 7 February 2022); ^+^ Water Resources Act and https://echa.europa.eu/documents/10162/13641/rest_microplastics_axvreport_annex_en.pdf/01741d07-f06b-bf32-8d6f-d6a8de54c4d0 (accessed on 7 February 2022). Administrative regulation on substances hazardous to water; https://echa.europa.eu/documents/10162/13641/rest_microplastics_axvreport_annex_en.pdf/01741d07-f06b-bf32-8d6f-d6a8de54c4d0 (accessed on 7 February 2022). # Directive (EU) 2020/2184 of the European Parliament and of the Council of 16 December 2020 on the quality of water intended for human consumption, https://eur-lex.europa.eu/legal-content/EN/TXT/PDF/?uri=CELEX:32020L2184&from=ES (accessed on 22 April 2022) [88]; ^X^ WFD Directive 2000/60/EC of the European Parliament and of the Council of 23 October 2000 establishing a framework for Community action in the field of water policy, https://eur-lex.europa.eu/resource.html?uri=cellar:5c835afb-2ec6-4577-bdf8-756d3d694eeb.0004.02/DOC_1&format=PDF (accessed on 22 April 2022).

**Table 2 sensors-22-04668-t002:** Spectral analysis of the diluted substances and their detection rate (* low; ** medium; *** high; - undetectable), measured by the spectrometer used.

Substance; CAS no.	Spectral Analysis		Detection	
	Raman	Fluorescence	Raman	Fluorescence
**Metformin hydrochloride**; 1115-70-4; purity > 99%	✓	✓	-	-
**Carbamazepine**; **297-46-4**; purity > 98%	✓	✓	**	*
**Hydrochlorothiazide**; 58-93-5; purity > 99%	✓	✓	*	**
**Acetaminophen**; 103-90-2; purity > 99%	✓	✓	**	*
**Naproxen**; 22204-53-1; purity > 98%	✓	✓	***	**
**Diclofenac**; 15307-93-4; purity > 98%	✓	✓	-	-
**Bromate**; 7789-38-0	✓	✓	-	**
**Bromide**; 7647-15-6	✓	✓	-	**
**Nitrate**; 7631-99-4	✓	✓	*	-
**Nitrite**; 7632-00-0	✓	✓	*	-
**Tryptophan**; 73-22-3	✓	✓	***	***
**Tyrosine**; 60-18-4	✓	✓	***	**
**DOC**	✕	✓	-	**

High detection rate down to 0.001 mg/L; medium detection rate down to 0.01 mg/L; low detection rate down to 0.1 mg/L. (✓) measured; (✕) not measured.

## Data Availability

Not applicable.

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
