# Peer review of "Possibilities of Real Time Monitoring of Micropollutants in Wastewater Using Laser-Induced Raman & Fluorescence Spectroscopy (LIRFS) and Artificial Intelligence (AI)"

_sensors, 2022, doi:10.3390/s22134668_

Round 1
Reviewer 1 Report
The manuscript "Possibilities of Real Time Monitoring of Micropollutants in Wastewater using Laser Induced Raman & Fluorescence Spectroscopy and Artificial Intelligence (AI) " by Claudia Post et al. tested the possibility of real time monitoring of micropollutants in wastewater using Laser Induced Raman and Fluorescence Spectroscopy in a combination with artificial intelligence approach using convolutional neural networks. This will be of great interest to readers in the corresponding field. I recommend the manuscript to be accepted for publication after minor revision. The issues are listed below
- Artificial intelligence is rarely introduced in the manuscript, how authors edit and use the artificial intelligence?
- In P4 L145, “…due to vibrational relaxation, also called internal conversion (blue arrows).” The vibrational relaxation and internal conversion are different, internal conversion is a non-radiative transition between different excited states, and vibrational relaxation is a non-radiative transition between different vibrational energy levels of the same excited state.
- When there are multiple pictures, they should be marked, such as (a), (b), (c)…. And in Fig.3, the cm-1 should be cm-1.
- In Fig.6, the scale bar in the upper left corner is not clear, some lens names are not marked.
- In Fig.10, what is marked by the green line in the middle? No obvious peaks are seen between 1300 cm-1 and 1400 cm-1. How to assign the peaks around 400 cm-1, 800 cm-1 and 2200 cm-1?
- In Fig.14, due to oversaturation of the CCD detector the 10 mg/L and 5 mg/L concentrated solution spectra are cut off, but why the detection threshold values of 10 mg/L and 5 mg/L are different?
- What is the different between Fig.19 and Fig.8?
- The authors showed the trend of photodegradation through 10 measurements, what is the test environment and whether it is affected by the time interval between each measurement?
Author Response
Please find response letter as word document.

Reviewer 2 Report
Referee’s Report on the paper:
Possibilities of Real Time Monitoring of Micropollutants in Wastewater using Laser Induced Raman & Fluorescence Spectroscopy (LIRFS) and Artificial Intelligence (AI)
By Claudia Post et al.
submitted to Sensors
General comments
The paper deals with some tests to verify if the Laser Induced Raman & Fluorescence Spectroscopy (LIRFS) could be a suitable method to individuate and evaluate micropollutants in waste water treatment plants (WWTP). Different micropollutants have been taken into account at different concentrations and their fluorescence and Raman spectra have been analysed.
This subject is potentially of interest for Sensors, but I’m not sure this is the most suitable Journal. In fact, the paper is based on the use of a commercial system and is very applicative. Sincerely, I would suggest the application to a Journal in the applicative field of water and waste treatment.
However, Editors will evaluate this consideration, in the meanwhile here below you can find my general and detailed comments that the Authors would take into account to revise the paper.
Generally speaking, I don’t like the look and the arrangement of the paper: is not in line with the title, sample and data presentation is not clear enough, too much importance is given to bibliography recalls.
Remarks
- Speaking about the general scheme of the paper, the paper seems to be composed by two different and unbalanced parts: before section 5 and from section 5.
The first one is for the most part is elementary and redundant information and bibliography. For example: figure 2 is, in my opinion, completely inappropriate. Stokes shift and Jablonski diagram can be recalled by a reference if necessary and moreover, in the present case, the result discussion does not refer strictly to such pieces of theory. Then their explanation seems to be unnecessary. The same for figure 3: they are literature figures taken from the web site of Photon Systems. Not only they are redundant but are also not clear, not homogeneous in the visual presentation and difficultly comparable (comments inside the boxes are not clear, x scale of the right figure is in part missing and unclear). The WWTP description is too detailed with respect to the experimental part (I would avoid to use a figure from the Encyclopaedia Britannica, figure 5).
The description of the commercial instrument is too long, too detailed, all the information can be easily found on the website of the company. Moreover, it is not clear if figure 7 is specifically produced for this paper or it is a literature figure previously produced by the company. In addition, the added value due to its presence is questionable.
In section 4, the software is described. Also in this case the novel Author contribution is not clear: In fact, the software seems to be commercial, some of the information are really elementary (i.e the standard deviation formula).
Also the novelty of figure 8 and figure 9 is not clear, since their raw data derive from literature. Maybe only the background subtraction is to refer the Authors.
The second part of the paper starts with section 5 (the first one because we have here two sections called 5).
This part includes the real paper, the experimental results and their discussion. In my opinion the paper could start here.
However also this part is mixed up and imprecise. I suggest to review it completely: a section on sample’s presentation is completely missing, they are vaguely cited inside the text. Figure 11 legend cites “extraction points” and at line 545 “field campaign “are cited, but all the results seem to refer to reference materials ad hoc diluted. The I suggest to produce a section devoted only to samples description and listing, maybe with a table to summarize which measurements were carried out on every sample (with dilution specification).
For some samples only fluorescence is presented for others also Raman spectrum. The real complementarity of the results of the two spectroscopic technique is not highlighted. The combination of the spectral features allows for unambiguous identification of all the analyzed substances? Or just of some of them? Or of none of them? Also here, I suggest to add a table clearly list results obtained with fluorescence and Raman for every sample. Moreover, I suggest to homogenize the visual presentation of the results.
- Nitrites and nitrites discussion: It seems very very similar to the one presented by the Photon Systems Company in its website: https://photonsystems.com/nitrate-and-nitrite-detection/
- Figure 25: figure end data here present in the box are from literature. Why to propose it stand alone? A sense can be found if reference spectrum was inserted in the right spectra of figure 24 to highlight the position of the peaks.
- I don’t understand the caption “raw data from [ref]” for almost all the experimental figures. Which is the processing applied to these data. As said, I don’t understand well, but I don’t like it.
- The role of “substance classification” is unclear. In section 4.2, from line 460 there is a brief description, in which 1700 training samples are cited. What they are? To classify what? The Authors say “23 substances”. Which? Which the connection with the data presented in section 5?
- Last but not least, the correlation with the title: here real time monitoring and artificial intelligence are cited. However, only lab tests were carried out, with samples containing just one substance at time (if differently conducted, then it is not clear), in ethanol and not in water. The real time measurements seem to be very far. Is the artificial intelligence used for data presented? Which is its role? From text seems none.
In conclusion, to try to save the paper I suggest to take out a large part of it and completely rearrange the rest, highlighting the novelty of the experimental data, in the present form not put in evidence.
Author Response

(The authors gave the same response as above.)

Reviewer 3 Report
The work used Laser Induced Raman and Fluorescence Spectroscopy (LIRFS) combined with an CNN based AI support to online monitor of micropollutants in wastewater. The study is novel, and the manuscript is mostly well-prepared. But the manuscript still needs major revisions before publication. Here are many specific comments:
- The left and right images in Figure 3 are inconsistent with the image description below figure 3.
- Although the literature related to deep UV is given in Section 2.1, the principle of separating Raman spectroscopy and fluorescence spectroscopy should also be explained in the manuscript.
- The CNN classification model mentioned in Section 4.2 should give a structure diagram.
- The 1700 spectral data mentioned in Section 4.2 are used to train the models, and the classification proportion of training set and test set is not given.
- What is a quartz glass cuvette and flow cell mentioned in Section 3.1? What's the difference?
- The specific information of peaks C mentioned in part 5. Results should be given, and the location should also be marked in Figure 12.
Author Response

(The authors gave the same response as above.)

Reviewer 4 Report
the authors present a fairly comprehensive manuscript dealing with the use of deep UV fluorescence in combination with Raman spectroscopy for the analysis of micropollutants in water. Their approach is quite original and could have a real impact on practice. The manuscript is written clearly and comprehensibly and without errors. The whole message of the manuscript is clearly presented, even with possible shortcomings of the demonstrated method. Therefore, I have not any serious comments on the work, just a few questions and comments that I would like to clarify before this manuscript can be published.
1) The authors use excitation at 248.6 nm. What about some other wavelengths? Can shift of the wavelength improve the signal for some analytes?
2) How the length of pulses is being selected? Empirically?
3) The authors mentioned microplastics as one of emerging pollutants. Would it be possible to use this method for detection/identification of this analyte?
4) Could this method, possibly, be used also for analysis of gaseous analytes?
5) "metforminhydrochlorid" should be spelled "metforminhydrochloride" throughout the manuscript
Author Response

(The authors gave the same response as above.)

Round 2
Reviewer 2 Report
I can notice an improvement in the quality paper.
But 2 points are still not accomplished:
- The complementarity of Raman and LIRFS on such kind of samples and
usefulness of using both of them combined together. It is not clear from
the text on which samples information from the two techniques have been
or can be combined and the originating benefit(In my opinion it could be
enough a comment and a table).
-The interferences between difference substances have not been studied.
Moreover, only pure substance not in water but in ethanol have been
studied (water used only in a few cases).
Author Response
Dear Reviewer 2, we are very grateful for your detailed work and followed your points:
Reviewer 2, Point 1: The complementarity of Raman and LIRFS on such kind of samples and usefulness of using both of them combined together. It is not clear from the text on which samples information from the two techniques have been or can be combined and the originating benefit(In my opinion it could be enough a comment and a table).
Response 1: We agree with you and added a table (Table 2 in line 750) to give the detection rates of all measured substances coming from the respective spectral analysis. For substances not detectable via Raman a fluorescence detection is very helpful (e.g., for bromate, bromide). For sum parameters like DOC sensitive fluorescence monitoring is very successful – if individual substances doesn’t need to be identified.
Reviewer 2, Point 2: The interferences between difference substances have not been studied. Moreover, only pure substance not in water but in ethanol have been studied (water used only in a few cases).
Response 2: To clarify this issue - 8 measured substances were diluted with Millipore® water and 4 with ethanol (compare with Table 1). We identified the characteristic Raman spectrum of every detectable substance with the pure powder (not possible for the salts but for the pharmaceuticals), but we need to dilute them to reach out for the lowest detectable signal (detection limit for the spectrometer used). This was possible with water for 8 substances but for the 4 pharmaceuticals only ethanol was solving.
Regarding interference, in the laboratory we only concentrated on the effect of the solvent (even water and ethanol) that manipulates the substance signal and tried to bypass it via software. But we only managed it properly regarding fluorescence measurements. You are completely right that WWTP samples contain of course remaining substances that interfere with our targets. Therefore, we used the collective samples stemming from the different treatment stages (Figure 9) and we could, for example detect nitrate even after biological treatment, but not bromat.
Reviewer 3 Report
I have no comments.
Author Response
Dear Reviewer 3,
we are very grateful for your detailed work and see that your recommendations helped to improve the comprehensibility of the paper.
with the best regards
Claudia Post